# What Can Be Learned by Knowing Only the Amino Acid Composition of Proteins?

**DOI:** 10.3390/ijms252413680

**Published:** 2024-12-21

**Authors:** Michail Yu. Lobanov, Alexey A. Surin, Oxana V. Galzitskaya

**Affiliations:** 1Institute of Protein Research, Russian Academy of Sciences, 142290 Pushchino, Russia; m.u.lobanov@mail.ru; 2Faculty of Informatics and Computer Engineering, MIREA—Russian Technological University, 119454 Moscow, Russia; alexey_junior02@mail.ru; 3Institute for Theoretical and Experimental Biophysics, Russian Academy of Sciences, 142290 Pushchino, Russia; 4Gamaleya Research Centre of Epidemiology and Microbiology, 123098 Moscow, Russia

**Keywords:** amino acid composition, protein, proteome, function, frequency, distribution

## Abstract

The amino acid composition of proteins depends on many factors. It varies in organisms that are distant in taxonomic position. The amino acid composition of proteins depends on the localization of proteins in cells and tissues and the structure of proteins. The question arises: is it possible to separate different proteomes using only the amino acid composition of proteins? Is it possible to determine, considering only its amino acid composition, to what structural class the protein under study will belong? We have developed a method and a measure that maximally separate two sets of proteins. As a result, we assign each protein an R-value, positive values of which are more characteristic of the first set, and negative ones—of the second. By studying the distribution of R in two sets, we can determine how much these sets differ in composition. Also, when examining a new protein, we can determine if it is more similar to the first set or the second. In this paper, we show that using only amino acid composition, it is possible to separate sets of proteins belonging to different organisms, as well as proteins that differ in function or structure. In all cases, we assign to proteins a measure R that maximally separates the studied sets. This approach can be further used to annotate proteins with unknown functions.

## 1. Introduction

The goal of recent works has been to find and define the structural and sequence features that are common to some class of proteins, for example, disordered or amyloidogenic regions [1,2,3,4,5,6,7,8]. Thanks to the wealth of data available in the Protein Data Bank, most analyses try and are able to discover common structural and chemical properties. AlphaFold has achieved the greatest success in this area [9]. It is reasonable to suppose that proteins grouped together on the basis of common architecture would reveal some commonality on the level of primary structure as well.

The amino acid composition of proteins is one of the most important parameters for determining the structure and function of proteins. It would seem to be a rather crude characteristic, but a large number of cysteines most likely indicate that these are keratins or keratin-associated proteins. Mucin-2 contains an abnormally large number of threonines (32%) and prolines (16%). Histone H1.5 consists of more than half alanines (27%) and prolines (29%). Even such a familiar protein as human beta-hemoglobin contains many histidines (12%) and valines (6%). The amino acid composition depends on the structure and function of the protein, as well as on the position of the organism on the evolutionary tree. We will consider this issue in more detail later.

Many protein structure prediction programs, both those with artificial intelligence and those using parameters/scales derived from the physical properties of amino acids, use the amino acid composition as one of the most important parameters [1,2,3,4]. Recently, we have developed a method that allows us to best separate peptides/proteins belonging to two different groups based only on their amino acid composition [10]. We have previously shown that this method can be used to predict antibacterial peptides, and this simple method works at the level of deep learning methods. We have learned to separate the amyloidogenic peptides from the non-amyloidogenic ones (the work has not been published yet). We want to extend our approach to whole proteins and observe what can be achieved using only amino acid composition without additional information. The goal of this work is to understand what features of proteins can be identified based only on their amino acid composition, and what is the scale of this tool in relation to such data sets as the proteome, individual protein sets such as ribosomal proteins, and proteins from different structural classes (α, β, α/β, α + β).

In this article, we will not study the question of the evolution of amino acids, although several interesting works are devoted to this matter [11,12]. By studying the evolution of some important protein folds, one can also follow changes in amino acid composition, and this was also the subject of many works [13,14]. Thus, it was shown that c class of proteins (α/β) is one of the most ancient [15] and more designable folds [16]. Statistical analysis shows that the four major structural classes of proteins (all-α, all-β, α/β, α + β) differ from each other in a statistically significant way in the number of rotatable angles φ, ψ, and χ, and the average number of contacts per residue. In general, among proteins of the same size, α/β proteins were shown to have, on average, a higher number of contacts per residue due to their more compact structure [17].

Studying the occurrence of homo-repeats for 20 amino acids, we realized that each proteome has its own prevailing type of homo-repeats [18]; for example, in humans, these are homo-repeats from prolines and glutamic acid [19]. In this paper, we want to understand what features of proteins can be identified taking into account only their amino acid composition. To see in which cases our approach allows us to predict something, and in which cases it does not. The latter is also interesting because it indicates the direction of the natural selection of amino acid composition for different tasks [20]. So far, we have been the first to consider the division of two sets of proteins according to their amino acid composition.

## 2. Results and Discussion

### 2.1. Features of Amino Acid Distribution in Proteins from Different Organisms

Each protein has its own ensemble of amino acid residues; some residues prevail in each protein compared to the proteomic values, and a set of amino acid residues that, in contrast, are few when compared again with the proteomic data. Usually, when comparing with proteomic values, the frequencies of occurrence for 20 amino acid residues are divided by the frequencies of occurrence in the proteome. These frequencies are called normalized frequencies, with the reference level being 1. Thus, for human myoglobin, the normalized frequencies for histidine and lysine are 2.2. times higher than in the proteomic data, and there is very little cysteine and arginine in this protein (Figure 1). At the same time, for such a common protein in the human body as actin, isoleucine and methionine prevail, at twice the level than in the proteome; also threonine and tyrosine, which are 1.5 times more in quantity. In human lysozyme, there are completely different amino acids: tryptophan (3 times more compared to the proteomic values), arginine (2 times more), cysteine (2.7 times more), and asparagine (2.2 times more). All this is encouraging, as it may be possible to separate two sets of proteins with different structures or functions, and also belonging to different organisms.

We selected seven proteomes with the highest percentage of reviewed proteins from different forms of life. For each proteome, we calculated the frequency of occurrence and the observed probability density for 20 amino acid residues. We introduced the concept of the observed probability density: *n*/(*N*∆). Here, *n* is the number of proteins in the selected range of amino acid frequencies, *N* is the total number of proteins in the proteome, and Δ is the interval width, which we considered as 0.01. With this width, the distribution looks smooth and without fluctuations. We showed that the distributions of 20 amino acid residues varied among organisms (Figure 2 and Appendix A).

A very common question is whether the occurrence of amino acids in proteins can be considered as a normal distribution. Here, we will analyze the distribution of amino acid frequencies in the human proteome. The proteome itself was taken from the UniProt database (only reviewed proteins were taken). A total of 20,360 proteins were analyzed. First of all, we would like to note the proteins that sharply stand out in amino acid frequencies. The Q156A1 or Ataxin-8 protein is remarkable in that it consists of only glutamines (79) and methionine at the N-terminus. F7VJQ1 or alternative prion protein contains 18 tryptophans with a protein size of only 73 amino acid residues. There are many keratin-binding proteins enriched with cysteine, tyrosine, and glycine. There are also other proteins that sharply stand out in frequencies. On the other hand, there are quite a few proteins that lack one or more amino acids (Figure 3).

1343, or 6.6%, of proteins do not contain tryptophan at all. Only 19, or 0.09%, of proteins do not contain serine, which emphasizes the importance of this amino acid for the formation of protein sequences. This distribution pattern is observed not only for the human proteome, but also for the other six proteomes we considered (see Appendix A). The proteins with the highest frequencies were analyzed above. Now let us analyze the typical pattern of amino acid occurrence in the human proteome (Figure 4).

The highest average frequency is for leucine (10.0%), while the lowest is for tryptophan (1.3%). The average frequency of 20 amino acids for seven proteomes is given in the Appendix A. It is important to note that for some amino acids, the average value is close to the standard deviation. This indicates the asymmetry of the frequency distribution. Let us examine the frequency distribution of specific amino acids and compare it with the normal distribution with the same average and standard deviation (SD).

For glycine, the mean is 0.067, and the standard deviation is 0.030 (Figure 5). Interestingly, the normal distribution does not fall to zero at zero frequency, although the mean is approximately twice the standard deviation. The maximum frequency of glycine is 0.55 for the protein-small cysteine and glycine repeat-containing protein 10 (A0A286YEX9 number in UniProtKB reviewed, MGCCGCGGCGGRCSGGCGGGCGGGCGGGCGGCGGGCGSYTTCR).

However, the frequency distribution of valine is practically no different from the normal one (the average is 0.060, the standard deviation is 0.020) (Figure 6). The example is also remarkable in that the average values for glycine and valine are close, but the distributions differ greatly.

The distributions for all amino acids in comparison with the normal distribution are presented in Figure 6. For some amino acids, the distributions are almost identical, while for others they are far from ideal. Thus, three groups can be distinguished. The first group is where the two distributions are almost identical. Such a distribution is typical for the following amino acids: I, V, L, N, and D. The second group includes those amino acids where the frequency at zero is visibly greater than zero: C, V, F, W, Y, N, H, and K. And finally, the third group is where there is obvious asymmetry: C, A, G, S, Q, E, R, H, K, and P.

Let us consider non-specific reasons why the distribution of amino acid frequencies differs from the normal distribution. First of all, it should be noted that it does not have to resemble the normal distribution. Firstly, we see asymmetry. This is due to the fact that the mean and standard deviation are close in magnitude. This means that the normal distribution inevitably goes beyond zero, while the frequency is by definition distributed from zero to one. The asymmetry coefficient is calculated using Equation (1).
(1)$$A=\frac{\left\langle {\left(x-\bar{x}\right)}^{3}\right\rangle }{\sigma^{3}}$$

Here, x is the variable parameter (in our case, the frequency of amino acid residues in proteins), and σ is the standard deviation. In this case, the maximum value of asymmetry (A) 6.69 is observed for cysteine (C) and the minimum of 0.25 for valine (V) for human proteome (for others, see the Appendix A). The average value of asymmetry is 1.6 for the human proteome. Such residues as L, V, N, and D have the lowest asymmetry (0.29, 0.25, 0.41, and 0.58, respectively) and a distribution close to the normal, for which the asymmetry is 0. Another reason is related to the sizes of proteins, which vary within a very wide range (Figure 7).

Let us assume that proteins do not differ in composition from random copolymers with amino acid residues. In this case, at a fixed size, we will observe a distribution close to normal. Let us consider an amino acid residue with a frequency of 0.05 = 1/20 and distributions for proteins of 100 and 400 amino acid residues (Figure 8). For different protein sizes, different normal distributions will ensue, and the combination of these distributions will differ from the normal distribution.

### 2.2. Separating Proteomes Using Amino Acid Composition Alone

As shown above, the frequencies of amino acids in proteins from different organisms vary greatly. This gives a reason to hope that by relying only on the amino acid composition, it will be possible to separate proteins from different organisms. We performed calculations for all possible pairs, and for each pair, we obtained a set of R-values for amino acid residues at which we can maximally separate proteins of these proteomes. The set of these R-values can be found in the Appendix A. The prediction accuracy and Z-score are presented in Table 1. It should be noted that the average protein size does not influence the performance of the method illustrated in Table 1. As we see from Table 1, the method can separate all pairs of organisms except humans and mice, which belong to the class of mammals. Most likely, the amino acid composition of proteins is too similar within the classes. However, in other cases, we correctly predict the separation of two sets of 74% to 88% of proteins.

The quality of separation of human and mouse proteomes is shown in Figure 9a. At Z = 0.14, there is practically no separation of sets. The r distributions are too similar to each other although we tried to make them as divergent as possible. If we exclude the human–mouse pair when comparing proteomes, Z varies from 0.81 to 1.53. Accordingly, balance accuracy (BA) varies from 74% to 88%. Interestingly, pairs of proteomes within the mammalian class are poorly separated, while eukaryotes with different bacteria are well separated. Moreover, the level of separation between bacteria is high and amounts to 84%.

How well can we separate proteomes in general? The answer to this question is given in Figure 10a. Each protein is predicted to belong to its own set or to another. Since we have seven proteomes, we can correctly predict the membership in six cases or none. All intermediate variants are also possible. For all but humans and mice, we can correctly assign a protein to its proteome in 60% of cases. As it is easy to understand, random guessing would work in only (1/2)^6^ cases, or 1.6%.

Another interesting set is ribosome proteins in bacterial proteomes. We selected four bacteria: *Thermus thermophilus* (*T. thermophilus*), *Staphylococcus aureus* (*S. aureus*), *Pseudomonas aeruginosa* (*P. aeruginosa*), and *Escherichia coli* (*E. coli*). We again considered all possible pairs and ran calculations using our method to find such R-values that separated the two sets as much as possible. Five pairs were separated with an accuracy from 87% to 94% or Z from 1.5 to 2.0 (Table 2). The exception was one pair, *P. aeruginosa* and *E. coli*, with BA = 68% and Z = 0.42. Characteristically, this pair belongs to one class, Gammaproteobacteria, while all other pairs diverge at the level following the superkingdom of bacteria: *P. aeruginosa* and *E. coli* belong to the kingdom Pseudomonadati, *S. aureus*—to the kingdom Bacillati, for *T. thermophilus* there is no kingdom, and the phylum Deinococcota is immediately after the superkingdom. As can be seen, the general principle is also preserved for bacteria: the greater the taxonomic relationship, the worse the protein sets are separated.

However, what happens if we try to separate the proteomes of whole bacteria? Calculations were made for pairs of proteomes, and R-values were again obtained (see Appendix A). Considering all proteins in the proteome leads to better separation of the protein sets from different organisms than when only ribosomal proteins are considered (Table 2). The pair *P. aeruginosa* and *E. coli* was still separated the worst, but already at the level of Z = 1.2 and BA = 82%. In the pair *T. thermophilus* and *S. aureus*, Z reached 3.2 and BA 99%. The quality of recognition of these protein sets is presented in Figure 10c,d. From 60% to 80% of proteins confidently correspond to their set.

We compared the R-values of these proteomes that separate the proteomes and those that separate only ribosomal proteins (Figure 11). The R-values are similar, but for some amino acids, they have different signs, such as threonine in the pair *T. thermophilus* and *P. aeruginosa*. For separating ribosomal proteins, the R-value of threonine is −1.77, and for whole proteomes it is 0.41. In different pairs, the correlation between R-values ranges from 28% to 61% (Figure 11). In this regard, the question arose whether it was possible to separate ribosomal proteins using the R-values obtained for whole proteomes. As it turned out, they are separated quite well, but worse than using the R-values optimal for separating ribosomal proteins. If we exclude the pair *P. aeruginosa* and *E. coli*, then BA varies from 69% to 80% and Z from 0.7 to 1.3.

### 2.3. Separation of Proteins with Different Gene Ontology Annotations

Any protein has a function and a localization in the cell and the body, and participates in various processes. This information is collected in Gene Ontology (GO) annotations for all well-studied proteins [21,22]. Is it possible to separate proteins with different cellular localization and different functions? To answer this question, we took proteins with five different GO annotations from the human proteome. We created a set of GO annotations so that no protein contained two annotations from our set. It should be noted that this is not the only possible set of non-overlapping GO annotations, but we left one as an example. The sets of proteins with different GO annotations were separated well. Z varied from 1.4 to 7.2, respectively, and BA from 89% to 100%, i.e., complete separation was achieved. The quality of recognition of its own set was above 80% (Figure 10b). Figure 9d shows an example of the separation of two sets.

### 2.4. Separation of Proteins into Structural Classes Using Only Amino Acid Composition

Above, we have shown that it is possible to separate proteomes if organisms belong to different classes. We have also shown that it is possible to separate proteins with different functions or localizations. Is it possible to separate sets that differ in protein structure? To answer this question, we took proteins belonging to different structural classes according to the SCOP 1.65 nomenclature: a—all α proteins, b—all β proteins, c—α/β proteins, and d—α + β proteins. The last two classes were separated with the greatest difficulty: Z = 0.5 and BA = 65%. This is not surprising, because the secondary structure in these classes is the same; it is just folded differently. For the remaining pairs, Z varied from 0.8 to 1.4, and BA from 71% to 85%. The quality of recognition of own class was at the level of 50% (Figure 10e). Note that random recognition would work at a level of 6%.

For α helical proteins, such amino acids as M, K, A, L, C, R, Q, E, and N are important for class separation. For β proteins—V, P, W, T, G, S, and N; for the c class of proteins—I, L, M, A, F, G, V, and Y; and for the d class of proteins—I, F, G, A, M, L, V, H, and Y. As we know, α helical proteins are enriched with such amino acids as hydrophobic and positively and negatively charged amino acids, which stabilize the dipole structure of the helix with their charges.

Lysine, arginine, cysteine, and glutamine with positive R-values are important only for the a class of proteins. For the b class of proteins, these are aromatic and polar amino acids for hydrogen bonding in β-sheets. Serine, threonine, and proline are important only for the b class of proteins. Isoleucine, phenylalanine, and tyrosine are important for c and d classes of proteins. Histidine is important for the d class of proteins.

## 3. Materials and Methods

### 3.1. Dataset of Proteomes

Seven proteomes with a high fraction of reviewed (Swiss-Prot) proteins: *Homo sapiens* (UP000005640, 20,360 proteins), *Mus musculus* (UP000000589, 17,179 proteins), *Drosophila melanogaster* (UP000000803, 3708 proteins), *Arabidopsis thaliana* (UP000006548, 16,298 proteins), *Bacillus subtilis* (UP000001570, 4191 proteins), *Escherichia coli* (UP000000625, 4401 proteins), *Thermus thermophilus* (UP000000532, 2227 proteins), *Staphylococcus aureus* (UP000008816, 2889 proteins), and *Pseudomonas aeruginosa* (UP000002438, 5563 proteins). Proteomes were taken from the UniProt database in December 2023.

### 3.2. Dataset of Proteins

Four sets of ribosomal proteins were taken from the proteomes: *Thermus thermophilus*, *Staphylococcus aureus*, *Pseudomonas aeruginosa*, and *Escherichia coli*. We used 53 proteins for *T. thermophilus* and 54 for the rest. Whole proteomes of these bacteria were also used (2227, 2889, 5563, and 4401 proteins, respectively; see dataset of proteomes).

Proteins from 4 main structural classes of SCOP 1.65 were as follows: class a (all-α proteins, 794 proteins), class b (all-β proteins, 928 proteins), class c (α/β proteins, 1089 proteins), and class d (958 proteins) [23]. Proteins were filtered to exclude those with identity greater than 25%.

We created five sets of proteins from the human proteome with different Gene Ontology (GO) annotations. The GO annotations themselves were selected according to the following principles: at least 100 proteins should have these annotations, and no protein should be included in two sets. It should be noted that by using these criteria, it was possible to select other sets of proteins with other annotations. Our method for generating a set of GO annotations was simple. We took one annotation and searched for all annotations that did not intersect with this one in the human proteome. If the selected annotation occurred in any protein with any of the previously selected ones, we discarded it. We considered annotations from the most common to the rarer ones. The information is summarized in Table 3.

### 3.3. The Algorithm of the Program for Separation

The algorithm for separating two sets of sequences consists of selecting 20 parameters (R-values) assigned to each amino acid. Knowing the amino acid composition of a protein or peptide, we can calculate the average value (r). If r > 0, we predict the protein as belonging to the first set; otherwise to the second. The algorithm itself is implemented on the website http://bioproteom.protres.ru/prod_scale/ (accessed on 1 November 2024) and is described in the paper [10].

When we have more than two sets of proteins, we consider *M* = *N* × (*N* − 1)/2 pairs of sets, where *N* is the number of sets. Accordingly, we get 20 × *M* R-values. For any protein from any set, we can make *N* − 1 predictions with our set and others. The results are shown in Figure 10, Figure 11 and Figure 12 and the Appendix A.

### 3.4. An Evaluation of the Quality of Prediction

As the main criterion for assessing the quality of prediction, we use balanced accuracy (BA). We have two sets of proteins, A and B. Within each set, there is a proportion of correctly predicted proteins *T*; then,
(2)$$BA=\left(T_{A}+T_{B}\right)/2$$

The main value to be maximized in our algorithm is Z:(3)$$Z=\frac{R_{A}-R_{B}}{{\left(S_{A}^{2}+S_{B}^{2}\right)}^{1/2}}$$

Here, R is the average r for the set, and S is the standard deviation. As is easy to see, Z will not change if we add any number to all R-values, nor if we multiply all R-values by any positive number. It is important to note that if the sets contain several thousand proteins, then at Z > 5 our BA becomes indistinguishable from 1; that is, the prediction becomes absolutely accurate. If the sets contain several hundred proteins, the prediction becomes absolutely accurate at Z > 4.

### 3.5. Parameters Used

We introduce the concept of the observed probability density: *n*/(*N*∆). Here, *n* is the number of proteins in the selected range of amino acid frequencies, *N* is the total number of proteins in the proteome, and Δ is the interval width.

Normalized frequency is the frequency of occurrence of an amino acid residue in a protein divided by the average frequency of occurrence in the proteome.

## 4. Conclusions

As can be seen from this work, our method allows both to estimate the degree of similarity of different sets of proteins and to predict their belonging to a particular set. It turned out that there are practically indistinguishable sets of proteins (human and mouse proteomes), and those separated quite confidently (GO:0004984 and GO:0019814 in the human proteome). The proposed method cannot separate sets of proteins with the same amino acid frequencies; that is, in closely related species, the frequencies are practically the same.

We evaluated the results of applying our method to different pairs of protein sets. Our method gives a line of divergence of two protein sets by amino acid composition and allows to separate protein sets by organisms, structure, position, and function of proteins. At 0.5 < Z < 1.5, the quality of separation increases linearly: balance accuracy reaches 90%, after which it very smoothly reaches 100%.

Our method detects the difference in amino acid frequencies. If the frequencies are the same in the two sets, then theoretically it should give Z = 0. In practice, it can catch and amplify the influence of fluctuations or random differences. Naturally, the larger the size of the proteins, the smaller the R fluctuation for protein sequences. The fewer protein sequences, the more fluctuations or random differences can be caught by our method. However, if the difference in frequencies is real, we will see it even on a small set of proteins. Theoretically, our method works if we have more than 19 sequences in both sets.

The main task that we want to address in the future is protein clustering. Usually, proteins are clustered using the alignment method, which clusters proteins by degree of relatedness. Our approach allows us to group non-homologous proteins into one group. With a high degree of probability, the localization and function of proteins should determine the general characteristics of the protein composition, even if they have different origins. The results of the work will allow us to annotate proteins for proteomes for which such a problem exists.

## Figures and Tables

**Figure 1 ijms-25-13680-f001:**
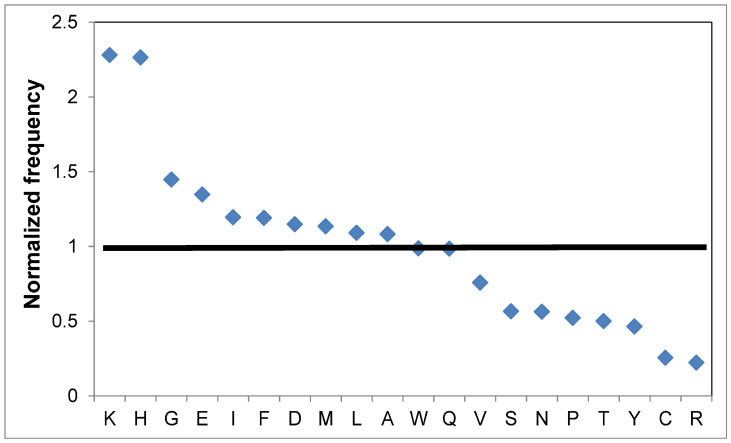
Amino acid abundances in human myoglobin normalized to human proteomic means with the reference level being 1.

**Figure 2 ijms-25-13680-f002:**
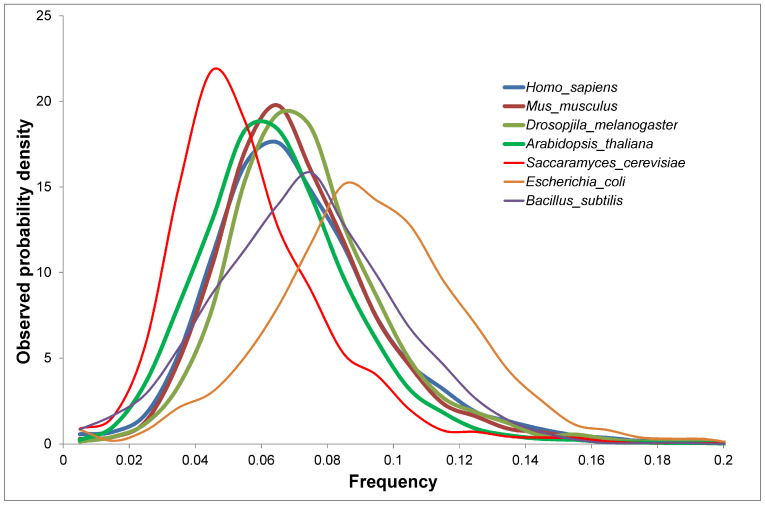
Distributions of alanine residues in seven different proteomes. The lowest frequency of alanine is found in yeast (0.057), and the highest in *E. coli* (0.093).

**Figure 3 ijms-25-13680-f003:**
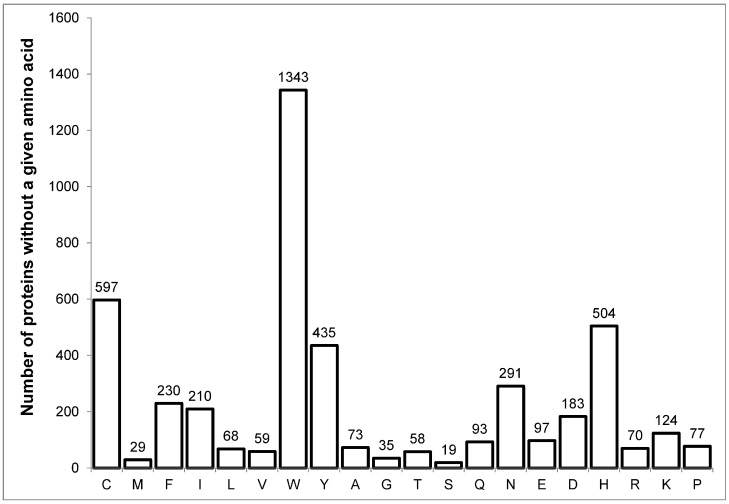
The number of proteins from the human proteome that do not contain a given amino acid.

**Figure 4 ijms-25-13680-f004:**
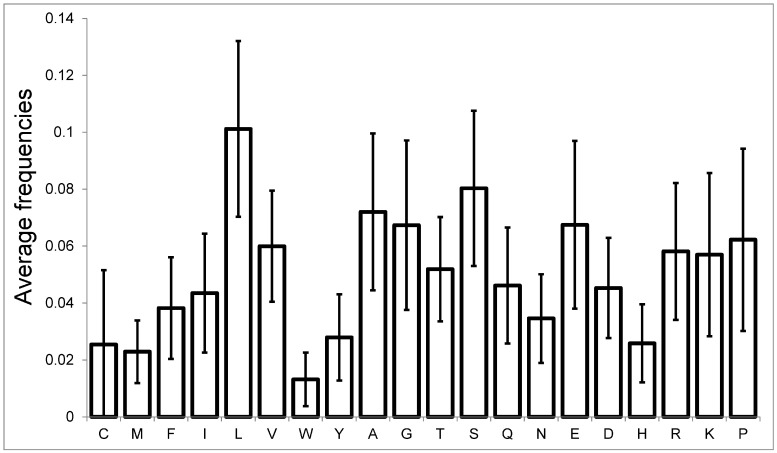
The frequencies and standard deviations for 20 amino acids in proteins from the human proteome.

**Figure 5 ijms-25-13680-f005:**
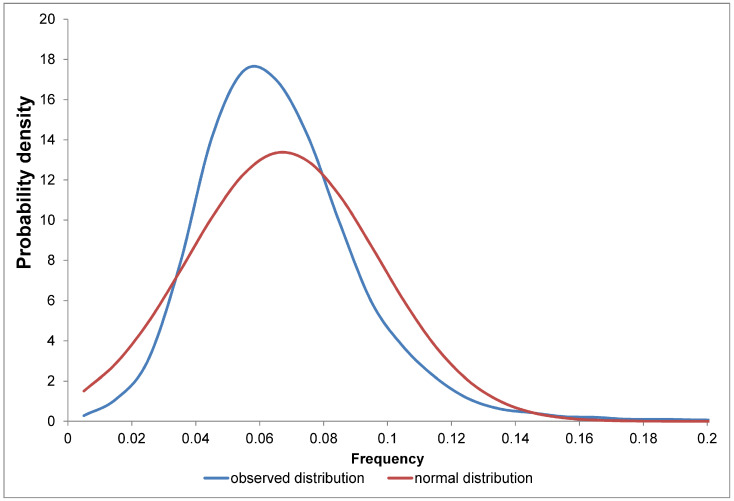
The probability density function for the occurrence of glycine in the human proteome compared to a normal distribution with the same mean and standard deviation.

**Figure 6 ijms-25-13680-f006:**
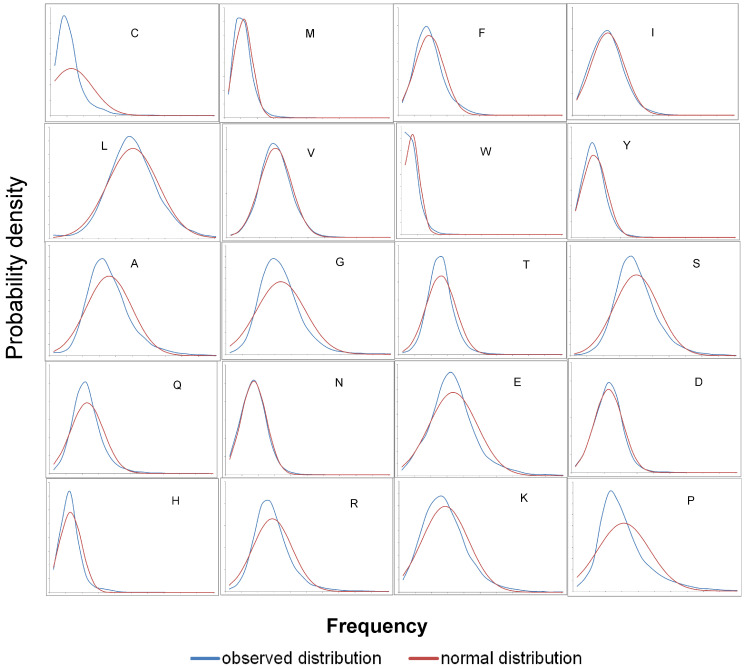
A comparison of the frequency distribution of 20 amino acids compared to the normal distribution for 20,360 proteins in the human proteome.

**Figure 7 ijms-25-13680-f007:**
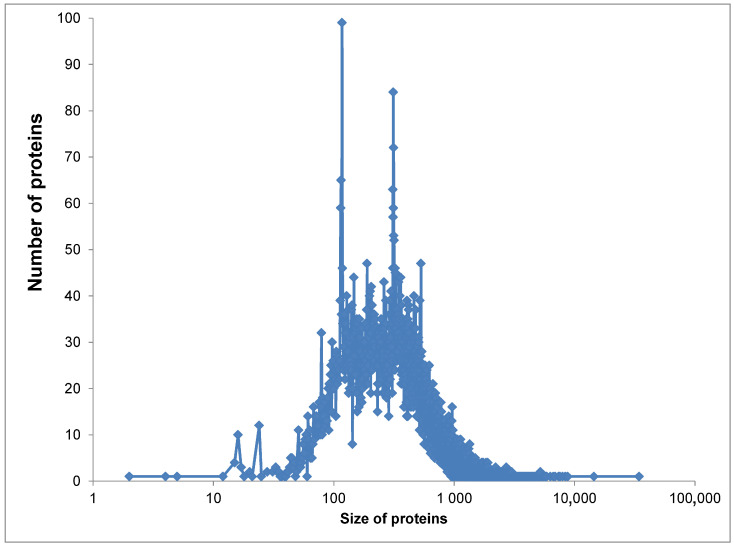
The distribution of the size (number of amino acid residues) of proteins in the human proteome.

**Figure 8 ijms-25-13680-f008:**
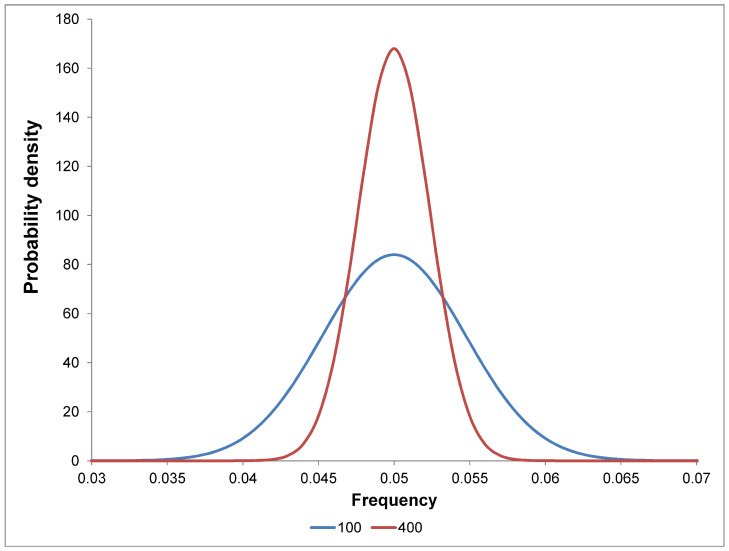
Probability density distributions of the occurrence of an amino acid with a frequency of 0.05 = 1/20 for proteins of 100 and 400 amino acid residues in size.

**Figure 9 ijms-25-13680-f009:**
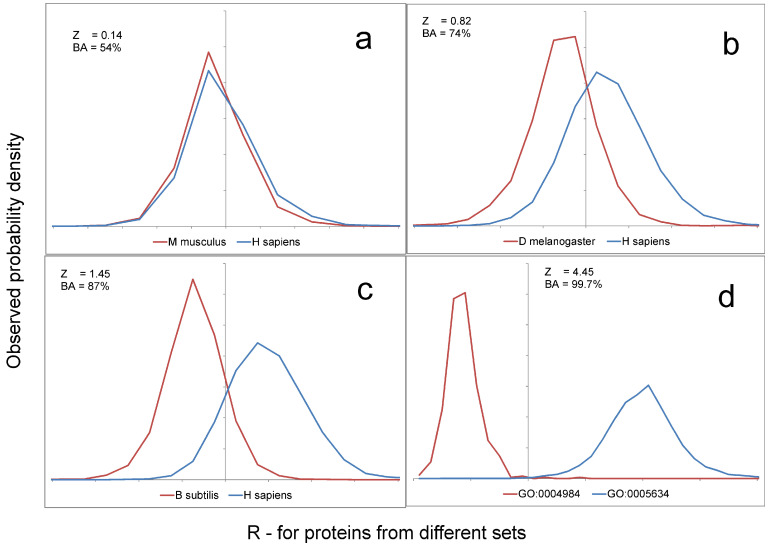
Distribution of R for proteins from different sets: (**a**) for *M. musculus* and *H. sapiens*; (**b**) for *D. melanogaster* and *H. sapiens*; (**c**) for *B. subtilis* and *H. sapiens*; (**d**) for GO:0004984 and GO:0005634. Examples with different Z are selected. Since we can multiply the R-value by any positive number, the scale does not matter.

**Figure 10 ijms-25-13680-f010:**
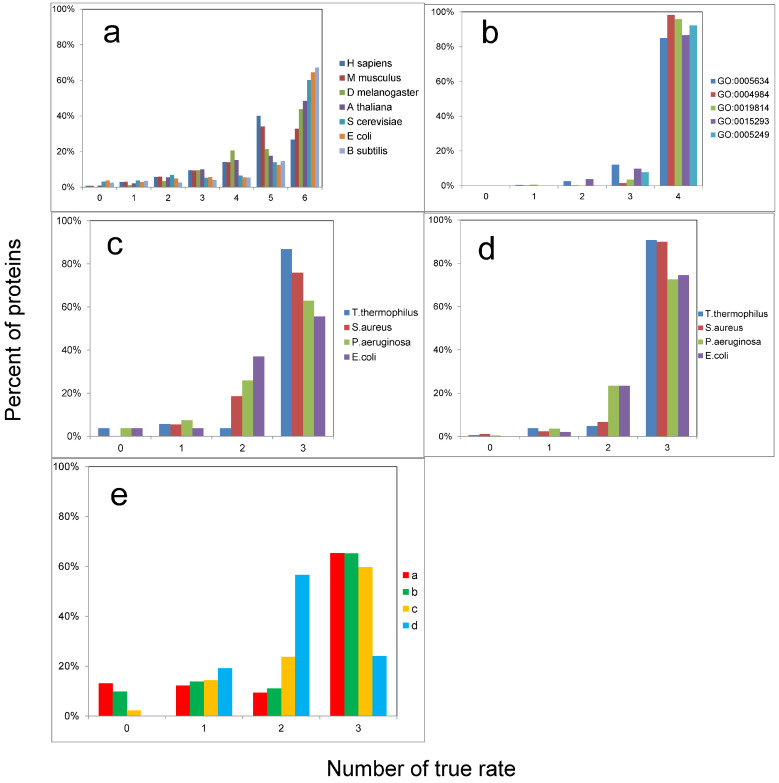
The number of correctly identified proteins from the sets taken: (**a**) seven proteomes; (**b**) five Gene Ontology (GO) annotations; (**c**) four sets of ribosomal proteins from bacterial proteomes; (**d**) four bacterial proteomes; (**e**) four structural classes from the SCOP 1.65 (a—all α proteins, b—all β proteins, c—α/β proteins, d—α + β proteins). For each protein in each set, there are *N* − 1 possible pairs. We can correctly identify a protein in all pairs (*N* − 1) or none (0).

**Figure 11 ijms-25-13680-f011:**
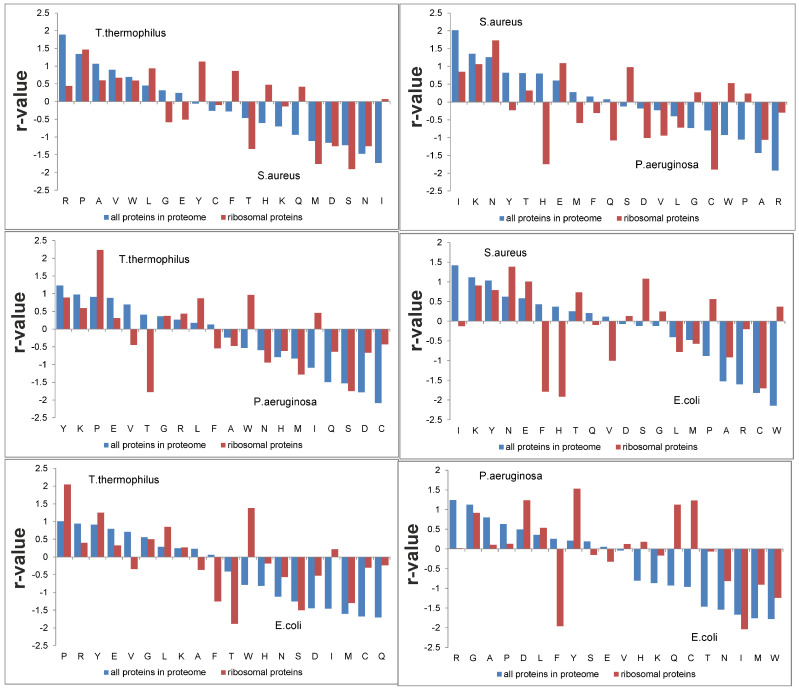
R-values for four sets of ribosomal proteins and whole bacterial proteomes containing these proteins.

**Figure 12 ijms-25-13680-f012:**
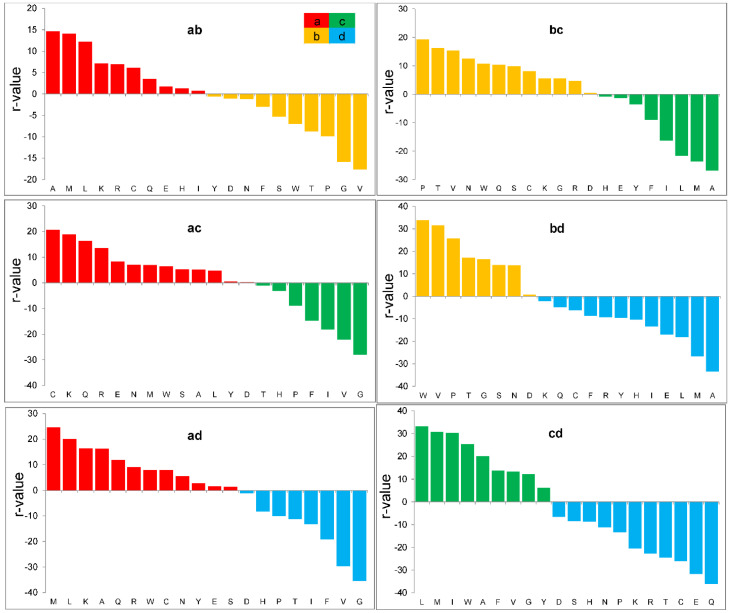
The R-values for pairs of protein sets from different structural classes from the SCOP 1.65 (a—all α proteins, b—all β proteins, c—α/β proteins, d—α + β proteins).

**Table 1 ijms-25-13680-t001:** Accuracy of separation of different pairs of proteomes. Above diagonal: balance accuracy (BA); below diagonal: Z.

Proteome	a	b	c	d	e	f	g
a, *H.sapiens*		54%	74%	76%	85%	85%	87%
b, *M.musculus*	0.14		74%	76%	85%	85%	88%
c, *D.melanogaster*	0.82	0.81		75%	80%	80%	85%
d, *A.thaliana*	0.91	0.90	0.83		80%	87%	86%
e, *S.cerevisiae*	1.31	1.36	1.07	1.06		88%	83%
f, *E.coli*	1.28	1.31	1.01	1.38	1.53		84%
g, *B.subtilis*	1.45	1.53	1.30	1.39	1.22	1.19	

**Table 2 ijms-25-13680-t002:** Accuracy of separation of different pairs of bacterial proteomes.

Proteome 1	Proteome 2	Z *	BA *	Z **	BA **	Z ***	BA ***
*T. thermophilus*	*S. aureus*	3.23	99.0%	2.00	94.4%	1.35	80.5%
*T. thermophilus*	*P. aeruginosa*	1.93	92.7%	1.64	88.8%	0.81	70.3%
*T. thermophilus*	*E. coli*	2.34	95.8%	1.72	89.7%	0.93	73.0%
*S. aureus*	*P. aeruginosa*	2.46	96.7%	1.56	87.0%	0.84	70.4%
*S. aureus*	*E. coli*	1.60	89.5%	1.50	90.7%	0.70	69.4%
*P. aeruginosa*	*E. coli*	1.18	82.1%	0.42	67.6%	0.15	57.4%

* Optimized and validated on whole proteomes. ** Optimized and tested on ribosomal proteins. *** Optimized on whole proteomes, validated on ribosomal proteins.

**Table 3 ijms-25-13680-t003:** Selected GO annotations and their description.

GO Annotation	Number of Proteins	Class	Description from UniProt
GO:0005634	9383	C	A membrane-bounded organelle of eukaryotic cells in which chromosomes are housed and replicated. In most cells, the nucleus contains all of the cell’s chromosomes except the organellar chromosomes and is the site of RNA synthesis and processing. In some species, or in specialized cell types, RNA metabolism or DNA replication may be absent.
GO:0004984	516	F	Combining with an odorant and transmitting the signal from one side of the membrane to the other to initiate a change in cell activity in response to detection of smell [24,25].
GO:0019814	170	C	A protein complex that in its canonical form is composed of two identical immunoglobulin heavy chains and two identical immunoglobulin light chains, held together by disulfide bonds and sometimes complexed with additional proteins. An immunoglobulin complex may be embedded in the plasma membrane or present in the extracellular space, in mucosal areas or other tissues, or circulating in the blood or lymph.
GO:0015293	134	F	Enables the active transport of a solute across a membrane by a mechanism whereby two or more species are transported together in the same direction in a tightly coupled process not directly linked to a form of energy other than chemiosmotic energy [26].
GO:0005249	129	F	Enables the transmembrane transfer of a potassium ion by a voltage-gated channel. A voltage-gated channel is a channel whose open state is dependent on the voltage across the membrane in which it is embedded.

## Data Availability

The data are contained within the article and Appendix A.

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
