# Peer review of "What Can Be Learned by Knowing Only the Amino Acid Composition of Proteins?"

_ijms, 2024, doi:10.3390/ijms252413680_

Round 1
Reviewer 1 Report
Comments and Suggestions for Authors
The study is interesting, as it introduces a classification method to distinguish proteins based on their amino acid compositions. The authors analyzed the distribution and frequencies of amino acids, providing insights that could help in differentiating proteins across various organisms or identifying differences in their structures and functions.
I have some recommendations that could help improve the manuscript:
Introduction:
- Please briefly include references to prior studies that used amino acid compositions for protein classification to provide additional context and background.
Results and Discussion:
- To highlight the novelty of your approach, compare your results with those from other similar studies.
- In Equation 1, please provide a detailed explanation of the variables and their meanings for better clarity.
- For Figure 7, consider separating the combined graphs into a more reader-friendly format. Additionally, the inner graph lacks axis labels—please include clear explanations for both the X and Y axes.
Materials and Methods:
- In the "Dataset of Proteomes" and "Dataset of Proteins" sections, specify the database(s) from which the datasets were obtained and include the date of download for transparency.
- Please spell out the full name of "GO" (Gene Ontology) the first time it appears in the text.
- When introducing Balanced Accuracy, include its abbreviation (BA) in parentheses for consistency.
- The conclusion should be improved to cover briefly the main points of study.
Reviewer 2 Report
Comments and Suggestions for Authors
The authors presented a method for estimating the similarities between sets of protein sequences based on only the amino acid composition. They demonstrated the proposed method's performance using a dataset of different species and human proteomes with different GO annotations. Their method is unique and has many potential applications. However, the manuscript in its current form is insufficient for publication. It is challenging to read and does not adhere to the standard format of scientific papers. Before submitting the manuscript to IJMS, I strongly recommend seeking feedback from other researchers to refine the content and improve clarity. Below are my detailed concerns:
Major:
1. In the abstract, “As previous studies have shown,…” and in the introduction, “The goal of last works…”. I do not know the authors' previous works. At least please cite your previous work.
2. L84 and 85, “** times more”. Which values does it discuss? Please clarify it.
3. Figure 1: please define “Frequency” and “Observed probability density”. When discussing new metrics or values, these terms must be defined clearly before explaining the results.
4. Figure 2: please define “Normalized frequency”.
5. L125: there is a definition of “Observed probability density”. Please move it to the paragraph that mentioned it the first time.
6. Figures 5 and 6: Please explain the values of the x-axis.
7. L159: I guess the maximum value corresponds to the asymmetry coefficient (A). Please clarify.
8. In section 2.2, please define BA and Z. Before discussing Table 1, please clearly explain how these values were computed. From this paragraph, I could not understand what the authors presented.
9. Figure 9: what is the “r-value”?
10. Figure 10: Please define the x-axis. Is it “Number of true rate”? What are 0, 1, 2, 3,..?
11. In conclusion, “Our approach can combine unrelated proteins into…”. I could not understand what the authors wanted to state here. Please clarify.
Comments on the Quality of English LanguageThe authors presented a method for estimating the similarities between sets of protein sequences based on only the amino acid composition. They demonstrated the proposed method's performance using a dataset of different species and human proteomes with different GO annotations. Their method is unique and has many potential applications. However, the manuscript in its current form is insufficient for publication. It is challenging to read and does not adhere to the standard format of scientific papers. Before submitting the manuscript to IJMS, I strongly recommend seeking feedback from other researchers to refine the content and improve clarity. Below are my detailed concerns:
Major:
1. In the abstract, “As previous studies have shown,…” and in the introduction, “The goal of last works…”. I do not know the authors' previous works. At least please cite your previous work.
2. L84 and 85, “** times more”. Which values does it discuss? Please clarify it.
3. Figure 1: please define “Frequency” and “Observed probability density”. When discussing new metrics or values, these terms must be defined clearly before explaining the results.
4. Figure 2: please define “Normalized frequency”.
5. L125: there is a definition of “Observed probability density”. Please move it to the paragraph that mentioned it the first time.
6. Figures 5 and 6: Please explain the values of the x-axis.
7. L159: I guess the maximum value corresponds to the asymmetry coefficient (A). Please clarify.
8. In section 2.2, please define BA and Z. Before discussing Table 1, please clearly explain how these values were computed. From this paragraph, I could not understand what the authors presented.
9. Figure 9: what is the “r-value”?
10. Figure 10: Please define the x-axis. Is it “Number of true rate”? What are 0, 1, 2, 3,..?
11. In conclusion, “Our approach can combine unrelated proteins into…”. I could not understand what the authors wanted to state here. Please clarify.
Round 2
Reviewer 2 Report
Comments and Suggestions for Authors
In this revision, the authors have significantly improved the clarity of the manuscript. Now, I can scientifically review the manuscript.
In my understanding, amino acid composition is a well-established method for characterizing the chemical properties and structural folds of proteins. The authors have demonstrated that amino acid composition can be effectively used to distinguish sequences based on organism type, structural classes, and GO terms. The data is well organized, but some additional discussions are necessary to clarify the results further. Below are my comments:
1. “2.2. Separating proteomes using amino acid composition alone.”: Please discuss why the proposed method could not separate human and mouse proteomes. Does it mean the proposed method can not separate these species that have close evolutionary relationships?
2. As the authors analyzed in Figures 7 and 8, the size of proteins affects probability density distribution. Does the average protein size influence the performance of the method in Table 1? For example, E. coli proteins are generally smaller than human proteins. Please clarify whether this observation affects the separation accuracy.
3. Since the number of proteins analyzed may vary across different species or GO annotations, please discuss how this variation impacts the results. Does the number of proteins in each dataset affect the separation performance?
4. In conclusion, please explain both the successes and limitations of using amino acid composition for sequence separation. Please consider the fundamental questions, such as why amino acid composition works effectively in some cases but fails in others (human vs. mouse). Is the performance correlated with factors such as protein size, the total number of proteins, or evolutionary distance? What are the unique advantages of the proposed method compared to existing techniques for sequence classification?
